# Stabilizing Off-Policy Q-Learning via Bootstrapping Error Reduction

**Aviral Kumar**[*]
UC Berkeley
aviralk@berkeley.edu

**Justin Fu**[*]
UC Berkeley
justinjfu@eecs.berkeley.edu

**George Tucker**
Google Brain
gjt@google.com

**Sergey Levine**
UC Berkeley, Google Brain
svlevine@eecs.berkeley.edu

## Abstract

Off-policy reinforcement learning aims to leverage experience collected from prior policies for sample-efficient learning. However, in practice, commonly used off-policy approximate dynamic programming methods based on Q-learning and actor-critic methods are highly sensitive to the data distribution, and can make only limited progress without collecting additional on-policy data. As a step towards more robust off-policy algorithms, we study the setting where the off-policy experience is fixed and there is no further interaction with the environment. We identify *bootstrapping error* as a key source of instability in current methods. Bootstrapping error is due to bootstrapping from actions that lie outside of the training data distribution, and it accumulates via the Bellman backup operator. We theoretically analyze bootstrapping error, and demonstrate how carefully constraining action selection in the backup can mitigate it. Based on our analysis, we propose a practical algorithm, bootstrapping error accumulation reduction (BEAR). We demonstrate that BEAR is able to learn robustly from different off-policy distributions, including random and suboptimal demonstrations, on a range of continuous control tasks.

## 1 Introduction

One of the primary drivers of the success of machine learning methods in open-world perception settings, such as computer vision [19] and NLP [8], has been the ability of high-capacity function approximators, such as deep neural networks, to learn generalizable models from large amounts of data. Reinforcement learning (RL) has proven comparatively difficult to scale to unstructured real-world settings because most RL algorithms require active data collection. As a result, RL algorithms can learn complex behaviors in simulation, where data collection is straightforward, but real-world performance is limited by the expense of active data collection. In some domains, such as autonomous driving [38] and recommender systems [3], previously collected datasets are plentiful. Algorithms that can utilize such datasets effectively would not only make real-world RL more practical, but also would enable substantially better generalization by incorporating diverse prior experience.

In principle, off-policy RL algorithms can leverage this data; however, in practice, off-policy algorithms are limited in their ability to learn entirely from off-policy data. Recent off-policy RL methods (e.g., [18, 29, 23, 9]) have demonstrated sample-efficient performance on complex tasks in robotics [23] and simulated environments [36]. However, these methods can still fail to learn when presented with arbitrary off-policy data without the opportunity to collect more experience

---

[*]Equal Contribution

from the environment. This issue persists even when the off-policy data comes from effective expert policies, which in principle should address any exploration challenge [6, 12, 11]. This sensitivity to the training data distribution is a limitation of practical off-policy RL algorithms, and one would hope that an off-policy algorithm should be able to learn reasonable policies through training on static datasets before being deployed in the real world.

In this paper, we aim to develop off-policy, value-based RL methods that can learn from large, static datasets. As we show, a crucial challenge in applying value-based methods to off-policy scenarios arises in the bootstrapping process employed when Q-functions are evaluated on out of *out-of-distribution* action inputs for computing the backup when training from off-policy data. This may introduce errors in the Q-function and the algorithm is unable to collect new data in order to remedy those errors, making training unstable and potentially diverging. Our primary contribution is an analysis of error accumulation in the bootstrapping process due to out-of-distribution inputs and a practical way of addressing this error. First, we formalize and analyze the reasons for instability and poor performance when learning from off-policy data. We show that, through careful action selection, error propagation through the Q-function can be mitigated. We then propose a principled algorithm called *bootstrapping error accumulation reduction* (BEAR) to control bootstrapping error in practice, which uses the notion of *support-set matching* to prevent error accumulation. Through systematic experiments, we show the effectiveness of our method on continuous-control MuJoCo tasks, with a variety of off-policy datasets: generated by a random, suboptimal, or optimal policies. BEAR is consistently robust to the training dataset, matching or exceeding the state-of-the-art in all cases, whereas existing algorithms only perform well for specific datasets.

## 2 Related Work

In this work, we study off-policy reinforcement learning with static datasets. Errors arising from inadequate sampling, distributional shift, and function approximation have been rigorously studied as "error propagation" in approximate dynamic programming (ADP) [4, 27, 10, 33]. These works often study how Bellman errors accumulate and propagate to nearby states via bootstrapping. In this work, we build upon tools from this analysis to show that performing Bellman backups on static datasets leads to error accumulation due to out-of-distribution values. Our approach is motivated as reducing the rate of propagation of error propagation between states.

Our approach constrains actor updates so that the actions remain in the support of the training dataset distribution. Several works have explored similar ideas in the context of off-policy learning learning in online settings. Kakade and Langford [22] shows that large policy updates can be destructive, and propose a conservative policy iteration scheme which constrains actor updates to be small for provably convergent learning. Grau-Moya et al. [16] use a learned prior over actions in the maximum entropy RL framework [25] and justify it as a regularizer based on mutual information. However, none of these methods use static datasets. Importance Sampling based distribution re-weighting [29, 15, 30, 26] has also been explored primarily in the context of off-policy policy evaluation.

Most closely related to our work is batch-constrained Q-learning (BCQ) [12] and SPIBB [24], which also discuss instability arising from previously unseen actions. Fujimoto et al. [12] show convergence properties of an action-constrained Bellman backup operator in tabular, error-free settings. We prove stronger results under approximation errors and provide a bound on the *suboptimality* of the solution. This is crucial as it drives the design choices for a practical algorithm. As a consequence, although we experimentally find that [12] outperforms standard Q-learning methods when the off-policy data is collected by an expert, BEAR outperforms [12] when the off-policy data is collected by a suboptimal policy, as is common in real-life applications. Empirically, we find BEAR achieves stronger and more consistent results than BCQ across a wide variety of datasets and environments. As we explain below, the BCQ constraint is too aggressive; BCQ generally fails to substantially improve over the behavior policy, while our method actually improves when the data collection policy is suboptimal or random. SPIBB [24], like BEAR, is an algorithm based on constraining the learned policy to the support of a behavior policy. However, the authors do not extend safe performance guarantees from the batch-constrained case to the relaxed support-constrained case, and do not evaluate on high-dimensional control tasks.

## 3 Background

We represent the environment as a Markov decision process (MDP) defined by a tuple $(\mathcal{S}, \mathcal{A}, P, R, \rho_0, \gamma)$, where $\mathcal{S}$ is the state space, $\mathcal{A}$ is the action space, $P(s'|s, a)$ is the transition

distribution, $\rho_0(s)$ is the initial state distribution, $R(s, a)$ is the reward function, and $\gamma \in (0, 1)$ is the discount factor. The goal in RL is to find a policy $\pi(a|s)$ that maximizes the expected cumulative discounted rewards which is also known as the return. The notation $\mu_\pi(s)$ denotes the discounted state marginal of a policy $\pi$, defined as the average state visited by the policy, $\sum_{t=0}^{\infty} \gamma^t p_\pi^t(s)$. $P^\pi$ is shorthand for the transition matrix from $s$ to $s'$ following a certain policy $\pi$, $p(s'|s) = E_\pi[p(s'|s, a)]$.

Q-learning learns the optimal state-action value function $Q^*(s, a)$, which represents the expected cumulative discounted reward starting in $s$ taking action $a$ and then acting optimally thereafter. The optimal policy can be recovered from $Q^*$ by choosing the maximizing action. Q-learning algorithms are based on iterating the Bellman optimality operator $\mathcal{T}$, defined as

$$(\mathcal{T}\hat{Q})(s, a) := R(s, a) + \gamma \mathbb{E}_{T(s'|s,a)}[\max_{a'} \hat{Q}(s', a')].$$

When the state space is large, we represent $\hat{Q}$ as a hypothesis from the set of function approximators $\mathcal{Q}$ (e.g., neural networks). In theory, the estimate of the $Q$-function is updated by projecting $\mathcal{T}\hat{Q}$ into $\mathcal{Q}$ (i.e., minimizing the mean squared Bellman error $\mathbb{E}_\nu[(Q - \mathcal{T}\hat{Q})^2]$, where $\nu$ is the state occupancy measure under the behaviour policy). This is also referred to a *Q-iteration*. In practice, an empirical estimate of $\mathcal{T}\hat{Q}$ is formed with samples, and treated as a supervised $\ell_2$ regression target to form the next approximate $Q$-function iterate. In large action spaces (e.g., continuous), the maximization $\max_{a'} Q(s', a')$ is generally intractable. Actor-critic methods [35, 13, 18] address this by additionally learning a policy $\pi_\theta$ that maximizes the $Q$-function. In this work, we study off-policy learning from a static dataset of transitions $\mathcal{D} = \{(s, a, s', R(s, a))\}$, collected under an unknown behavior policy $\beta(\cdot|s)$. We denote the distribution over states and actions induced by $\beta$ as $\mu(s, a)$.

## 4    Out-of-Distribution Actions in Q-Learning

Q-learning methods often fail to learn on static, off-policy data, as shown in Figure 1. At first glance, this resembles overfitting, but increasing the size of the static dataset does not rectify the problem, suggesting the issue is more complex. We can understand the source of this instability by examining the form of the Bellman backup. Although minimizing the mean squared Bellman error corresponds to a supervised regression problem, the targets for this regression are themselves derived from the current Q-function estimate. The targets are calculated by maximizing the learned $Q$-values with respect to the action at the next state. However, the $Q$-function estimator is only reliable on inputs from the same

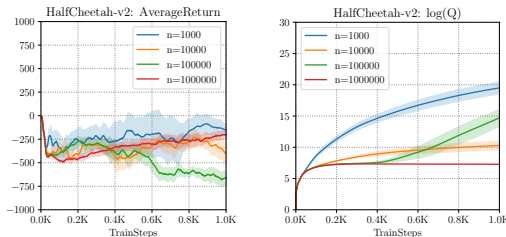

Figure 1: Performance of SAC on HalfCheetah-v2 (return (left) and $\log$ Q-values (right)) with off-policy expert data w.r.t. number of training samples ($n$). Note the large discrepancy between returns (which are negative) and $\log$ Q-values (which have large positive values), which is not solved with additional samples.

distribution as its training set. As a result, naïvely maximizing the value may evaluate the $\hat{Q}$ estimator on actions that lie far outside of the training distribution, resulting in pathological values that incur large error. We refer to these actions as out-of-distribution (OOD) actions.

Formally, let $\zeta_k(s, a) = |Q_k(s, a) - Q^*(s, a)|$ denote the total error at iteration $k$ of Q-learning, and let $\delta_k(s, a) = |Q_k(s, a) - \mathcal{T}Q_{k-1}(s, a)|$ denote the current Bellman error. Then, we have $\zeta_k(s, a) \leq \delta_k(s, a) + \gamma \max_{a'} \mathbb{E}_{s'}[\zeta_{k-1}(s', a')]$. In other words, errors from $(s', a')$ are discounted, then accumulated with new errors $\delta_k(s, a)$ from the current iteration. We expect $\delta_k(s, a)$ to be high on OOD states and actions, as errors at these state-actions are never directly minimized while training.

To mitigate bootstrapping error, we can restrict the policy to ensure that it output actions that lie in the support of the training distribution. This is distinct from previous work (e.g., BCQ [12]) which implicitly constrains the *distribution* of the learned policy to be close to the behavior policy, similarly to behavioral cloning [31]. While this is sufficient to ensure that actions lie in the training set with high probability, it is overly restrictive. For example, if the behavior policy is close to uniform, the learned policy will behave randomly, resulting in poor performance, even when the data is sufficient to learn a strong policy (see Figure 2 for an illustration). Formally, this means that a learned policy $\pi(a|s)$ has positive density *only where* the density of the behaviour policy $\beta(a|s)$ is more than a threshold (i.e., $\forall a, \beta(a|s) \leq \varepsilon \implies \pi(a|s) = 0$), instead of a closeness constraint on the value of

the density $\pi(a|s)$ and $\beta(a|s)$. Our analysis instead reveals a tradeoff between staying within the data distribution and finding a suboptimal solution when the constraint is too restrictive. Our analysis motivates us to restrict the support of the learned policy, but not the probabilities of the actions lying within the support. This avoids evaluating the Q-function estimator on OOD actions, but remains flexible in order to find a performant policy. Our proposed algorithm leverages this insight.

## 4.1 Distribution-Constrained Backups

In this section, we define and analyze a backup operator that restricts the set of policies used in the maximization of the Q-function, and we derive performance bounds which depend on the restricted set. This provides motivation for constraining policy support to the data distribution. We begin with the definition of a distribution-constrained operator:

**Definition 4.1** (Distribution-constrained operators). *Given a set of policies $\Pi$, the distribution-constrained backup operator is defined as:*

$$\mathcal{T}^{\Pi}Q(s,a) \stackrel{\text{def}}{=} \mathbb{E}\left[R(s,a) + \gamma \max_{\pi \in \Pi} \mathbb{E}_{P(s'|s,a)}\left[V(s')\right]\right] \qquad V(s) \stackrel{\text{def}}{=} \max_{\pi \in \Pi} \mathbb{E}_{\pi}[Q(s,a)] \ .$$

This backup operator satisfies properties of the standard Bellman backup, such as convergence to a fixed point, as discussed in Appendix A. To analyze the (sub)optimality of performing this backup under approximation error, we first quantify two sources of error. The first is a *suboptimality bias*. The optimal policy may lie outside the policy constraint set, and thus a suboptimal solution will be found. The second arises from distribution shift between the training distribution and the policies used for backups. This formalizes the notion of OOD actions. To capture suboptimality in the final solution, we define a *suboptimality constant*, which measures how far $\pi^*$ is from $\Pi$.

**Definition 4.2** (Suboptimality constant). *The suboptimality constant is defined as:*

$$\alpha(\Pi) = \max_{s,a} |\mathcal{T}^{\Pi}Q^*(s,a) - \mathcal{T}Q^*(s,a)|.$$

Next, we define a concentrability coefficient [28], which quantifies how far the visitation distribution generated by policies from $\Pi$ is from the training data distribution. This constant captures the degree to which states and actions are out of distribution.

**Assumption 4.1** (Concentrability). *Let $\rho_0$ denote the initial state distribution, and $\mu(s,a)$ denote the distribution of the training data over $\mathcal{S} \times \mathcal{A}$, with marginal $\mu(s)$ over $\mathcal{S}$. Suppose there exist coefficients $c(k)$ such that for any $\pi_1, ...\pi_k \in \Pi$ and $s \in \mathcal{S}$:*

$$\rho_0 P^{\pi_1} P^{\pi_2} ... P^{\pi_k}(s) \leq c(k)\mu(s),$$

*where $P^{\pi_i}$ is the transition operator on states induced by $\pi_i$. Then, define the concentrability coefficient $C(\Pi)$ as*

$$C(\Pi) \stackrel{\text{def}}{=} (1-\gamma)^2 \sum_{k=1}^{\infty} k\gamma^{k-1}c(k).$$

To provide some intuition for $C(\Pi)$, if $\mu$ was generated by a single policy $\pi$, and $\Pi = \{\pi\}$ was a singleton set, then we would have $C(\Pi) = 1$, which is the smallest possible value. However, if $\Pi$ contained policies far from $\pi$, the value could be large, potentially infinite if the support of $\Pi$ is not contained in $\pi$. Now, we bound the performance of approximate distribution-constrained Q-iteration:

**Theorem 4.1.** *Suppose we run approximate distribution-constrained value iteration with a set constrained backup $\mathcal{T}^{\Pi}$. Assume that $\delta(s,a) \geq \max_k |Q_k(s,a) - \mathcal{T}^{\Pi}Q_{k-1}(s,a)|$ bounds the Bellman error. Then,*

$$\lim_{k \to \infty} \mathbb{E}_{\rho_0}[|V^{\pi_k}(s) - V^*(s)|] \leq \frac{\gamma}{(1-\gamma)^2}\left[C(\Pi)\mathbb{E}_{\mu}[\max_{\pi \in \Pi}\mathbb{E}_{\pi}[\delta(s,a)]] + \frac{1-\gamma}{\gamma}\alpha(\Pi)\right]$$

*Proof.* See Appendix B, Theorem B.1 □

This bound formalizes the tradeoff between keeping policies chosen during backups close to the data (captured by $C(\Pi)$) and keeping the set $\Pi$ large enough to capture well-performing policies (captured by $\alpha(\Pi)$). When we expand the set of policies $\Pi$, we are increasing $C(\Pi)$ but decreasing $\alpha(\Pi)$. An example of this tradeoff, and how a careful choice of $\Pi$ can yield superior results, is given in a tabular

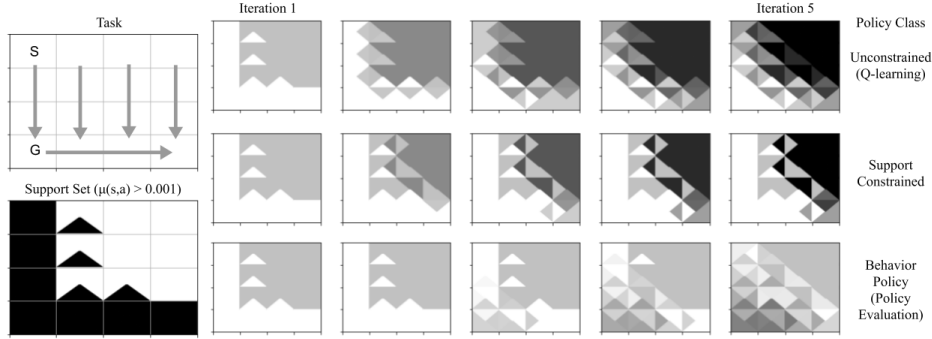

Figure 2: Visualized error propagation in Q-learning for various choices of the constraint set $\Pi$: unconstrained (top row) distribution-constrained (middle), and constrained to the behaviour policy (policy-evaluation, bottom). Triangles represent Q-values for actions that move in different directions. The task (left) is to reach the bottom-left corner (G) from the top-left (S), but the behaviour policy (visualized as arrows in the task image, support state-action pairs are shown in black on the support set image) travels to the bottom-right with a small amount of $\epsilon$-greedy exploration. Dark values indicate high error, and light values indicate low error. Standard backups propagate large errors from the low-support regions into the high-support regions, leading to high error. Policy evaluation reduces error propagation from low-support regions, but introduces significant suboptimality bias, as the data policy is not optimal. A carefully chosen distribution-constrained backup strikes a balance between these two extremes, by confining error propagation in the low-support region while introducing minimal suboptimality bias.

gridworld example in Fig. 2, where we visualize errors accumulated during distribution-constrained Q-iteration for different choices of $\Pi$.

Finally, we motivate the use of support sets to construct $\Pi$. We are interested in the case where $\Pi_\epsilon = \{\pi \mid \pi(a|s) = 0 \text{ whenever } \beta(a|s) < \epsilon\}$, where $\beta$ is the behavior policy (i.e., $\Pi$ is the set of policies that have support in the probable regions of the behavior policy). Defining $\Pi_\epsilon$ in this way allows us to bound the concentrability coefficient:

**Theorem 4.2.** *Assume the data distribution $\mu$ is generated by a behavior policy $\beta$. Let $\mu(s)$ be the marginal state distribution under the data distribution. Define $\Pi_\epsilon = \{\pi \mid \pi(a|s) = 0 \text{ whenever } \beta(a|s) < \epsilon\}$ and let $\mu_{\Pi_\epsilon}$ be the highest discounted marginal state distribution starting from the initial state distribution $\rho$ and following policies $\pi \in \Pi_\epsilon$ at each time step thereafter. Then, there exists a concentrability coefficient $C(\Pi_\epsilon)$ which is bounded:*

$$C(\Pi_\epsilon) \leq C(\beta) \cdot \left(1 + \frac{\gamma}{(1-\gamma)f(\epsilon)}(1-\epsilon)\right)$$

*where $f(\epsilon) \overset{\text{def}}{=} \min_{s \in \mathcal{S}, \mu_{\Pi_\epsilon}(s) > 0}[\mu(s)] > 0$.*

*Proof.* See Appendix B, Theorem B.2 □

Qualitatively, $f(\epsilon)$ is the minimum discounted visitation marginal of a state under the behaviour policy if only actions which are more than $\epsilon$ likely are executed in the environment. Thus, using support sets gives us a single lever, $\epsilon$, which simultaneously trades off the value of $C(\Pi)$ and $\alpha(\Pi)$. Not only can we provide theoretical guarantees, we will see in our experiments (Sec. 6) that constructing $\Pi$ in this way provides a simple and effective method for implementing distribution-constrained algorithms.

Intuitively, this means we can prevent an increase in overall error in the Q-estimate by selecting policies supported on the support of the training action distribution, which would ensure roughly bounded projection error $\delta_k(s, a)$ while reducing the suboptimality bias, potentially by a large amount. Bounded error $\delta_k(s, a)$ on the support set of the training distribution is a reasonable assumption when using highly expressive function approximators, such as deep networks, especially if we are willing to reweight the transition set [32, 11]. We further elaborate on this point in Appendix C.

# 5 Bootstrapping Error Accumulation Reduction (BEAR)

We now propose a practical actor-critic algorithm (built on the framework of TD3 [13] or SAC [18]) that uses distribution-constrained backups to reduce accumulation of bootstrapping error. The key insight is that we can search for a policy with the same support as the training distribution, while

preventing accidental error accumulation. Our algorithm has two main components. Analogous to BCQ [13], we use $K$ Q-functions and use the minimum Q-value for policy improvement, and design a constraint which will be used for searching over the set of policies $\Pi_\epsilon$, which share the same support as the behaviour policy. Both of these components will appear as modifications of the policy improvement step in actor-critic style algorithms. We also note that policy improvement can be performed with the mean of the K Q-functions, and we found that this scheme works as good in our experiments.

We denote the set of Q-functions as: $\hat{Q}_1, \cdots, \hat{Q}_K$. Then, the policy is updated to maximize the conservative estimate of the Q-values within $\Pi_\epsilon$:

$$\pi_\phi(s) := \max_{\pi \in \Pi_\epsilon} \mathbb{E}_{a \sim \pi(\cdot|s)} \left[ \min_{j=1,..,K} \hat{Q}_j(s,a) \right]$$

In practice, the behaviour policy $\beta$ is unknown, so we need an approximate way to constrain $\pi$ to $\Pi$. We define a differentiable constraint that approximately constrains $\pi$ to $\Pi$, and then approximately solve the constrained optimization problem via dual gradient descent. We use the sampled version of maximum mean discrepancy (MMD) [17] between the unknown behaviour policy $\beta$ and the actor $\pi$ because it can be estimated based solely on samples from the distributions. Given samples $x_1, \cdots, x_n \sim P$ and $y_1, \cdots, y_m \sim Q$, the sampled MMD between $P$ and $Q$ is given by:

$$\text{MMD}^2(\{x_1, \cdots, x_n\}, \{y_1, \cdots, y_m\}) = \frac{1}{n^2} \sum_{i,i'} k(x_i, x_{i'}) - \frac{2}{nm} \sum_{i,j} k(x_i, y_j) + \frac{1}{m^2} \sum_{j,j'} k(y_j, y_{j'}).$$

Here, $k(\cdot, \cdot)$ is any universal kernel. In our experiments, we find both Laplacian and Gaussian kernels work well. The expression for MMD does not involve the density of either distribution and it can be optimized directly through samples. Empirically we find that, in the low-intermediate sample regime, the sampled MMD between $P$ and $Q$ is similar to the MMD between a uniform distribution over $P$'s support and $Q$, which makes MMD roughly suited for constraining distributions to a given support set. (See Appendix C.3 for numerical simulations justifying this approach).

Putting everything together, the optimization problem in the policy improvement step is

$$\pi_\phi := \max_{\pi \in \Delta_{|S|}} \mathbb{E}_{s \sim \mathcal{D}} \mathbb{E}_{a \sim \pi(\cdot|s)} \left[ \min_{j=1,..,K} \hat{Q}_j(s,a) \right] \quad \text{s.t.} \quad \mathbb{E}_{s \sim \mathcal{D}}[\text{MMD}(\mathcal{D}(s), \pi(\cdot|s))] \leq \varepsilon \quad (1)$$

where $\varepsilon$ is an approximately chosen threshold. We choose a threshold of $\varepsilon = 0.05$ in our experiments. The algorithm is summarized in Algorithm 1.

How does BEAR connect with distribution-constrained backups described in Section 4.1? Step 5 of the algorithm restricts $\pi_\phi$ to lie in the support of $\beta$. This insight is formally justified in Theorems 4.1 & 4.2 ($C(\Pi_\varepsilon)$ is bounded). Computing distribution-constrained backup exactly by maximizing over $\pi \in \Pi_\varepsilon$ is intractable in practice. As an approximation, we sample Dirac policies in the support of $\beta$ (Alg 1, Line 5) and perform empirical maximization to compute the backup. As the maximization is performed over a *narrower* set of Dirac policies ($\{\delta_{a_i}\} \subseteq \Pi_\varepsilon$), the bound in Theorem 4.1 still holds. Empirically, we show in Section 6 that this approximation is sufficient to outperform previous methods. This connection is briefly discussed in Appendix C.2.

---

**Algorithm 1** BEAR Q-Learning (BEAR-QL)

---

**input** : Dataset $\mathcal{D}$, target network update rate $\tau$, mini-batch size $N$, sampled actions for MMD $n$, minimum $\lambda$

1: Initialize Q-ensemble $\{Q_{\theta_i}\}_{i=1}^K$, actor $\pi_\phi$, Lagrange multiplier $\alpha$, target networks $\{Q_{\theta_i'}\}_{i=1}^K$, and a target actor $\pi_{\phi'}$, with $\phi' \leftarrow \phi, \theta_i' \leftarrow \theta_i$
2: **for** $t$ in $\{1, \ldots, N\}$ **do**
3:      Sample mini-batch of transitions $(s, a, r, s') \sim \mathcal{D}$
     **Q-update:**
4:      Sample $p$ action samples, $\{a_i \sim \pi_{\phi'}(\cdot|s')\}_{i=1}^p$
5:      Define $y(s,a) := \max_{a_i}[\lambda \min_{j=1,..,K} Q_{\theta_j'}(s', a_i) + (1-\lambda) \max_{j=1,..,K} Q_{\theta_j'}(s', a_i)]$
6:      $\forall i, \theta_i \leftarrow \arg\min_{\theta_i}(Q_{\theta_i}(s,a) - (r + \gamma y(s,a)))^2$
     **Policy-update:**
7:      Sample actions $\{\hat{a}_i \sim \pi_\phi(\cdot|s)\}_{i=1}^m$ and $\{a_j \sim \mathcal{D}(s)\}_{j=1}^n$, $n$ preferably an intermediate integer(1-10)
8:      Update $\phi, \alpha$ by minimizing Equation 1 by using dual gradient descent with Lagrange multiplier $\alpha$
9:      **Update Target Networks:** $\theta_i' \leftarrow \tau\theta_i + (1-\tau)\theta_i'$; $\phi' \leftarrow \tau\phi + (1-\tau)\phi'$
10: **end for**

---

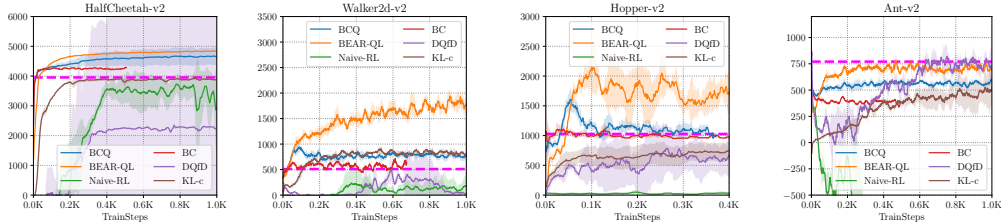

Figure 3: Average performance of BEAR-QL, BCQ, Naïve RL and BC on medium-quality data averaged over 5 seeds. BEAR-QL outperforms both BCQ and Naïve RL. Average return over the training data is indicated by the magenta line. One step on the x-axis corresponds to 1000 gradient steps.

In summary, the actor is updated towards maximizing the Q-function while still being constrained to remain in the valid search space defined by $\Pi_\epsilon$. The Q-function uses actions sampled from the actor to then perform distribution-constrained Q-learning, over a reduced set of policies. At test time, we sample $p$ actions from $\pi_\phi(s)$ and the Q-value maximizing action out of these is executed in the environment. Implementation and other details are present in Appendix D.

## 6   Experiments

In our experiments, we study how BEAR performs when learning from static off-policy data on a variety of continuous control benchmark tasks. We evaluate our algorithm in three settings: when the dataset $\mathcal{D}$ is generated by **(1)** a completely random behaviour policy, **(2)** a partially trained, medium scoring policy, and **(3)** an optimal policy. Condition **(2)** is of particular interest, as it captures many common use-cases in practice, such as learning from imperfect demonstration data (e.g., of the sort that are commonly available for autonomous driving [14]), or reusing previously collected experience during off-policy RL. We compare our method to several prior methods: a baseline actor-critic algorithm (TD3), the BCQ algorithm [12], which aims to address a similar problem, as discussed in Section 4, KL-control [21] (which solves a KL-penalized RL problem similarly to maximum entropy RL), a static version of DQfD [20] (where a constraint to upweight Q-values of state-action pairs observed in the dataset is added as an auxiliary loss on top a regular actor-critic algorithm), and a behaviour cloning (BC) baseline, which simply imitates the data distribution. This serves to measure whether each method actually performs effective RL, or simply copies the data. We report the average evaluation return over 5 seeds of the policy given by the learned algorithm, in the form of a learning curve as a function of number of gradient steps taken by the algorithm. These samples are only collected for evaluation, and are not used for training.

### 6.1   Performance on Medium-Quality Data

We first discuss the evaluation of condition with "mediocre" data **(2)**, as this condition resembles the settings where we expect training on offline data to be most useful. We collected one million transitions from a partially trained policy, so as to simulate imperfect demonstration data or data from a mediocre prior policy. In this scenario, we found that BEAR-QL consistently outperforms both BCQ [12] and a naïve off-policy RL baseline (TD3) by large margins, as shown in Figure 3. This scenario is the most relevant from an application point of view, as access to optimal data may not be feasible, and random data might have inadequate exploration to efficient learn a good policy. We also evaluate the accuracy with which the learned Q-functions predict actual policy returns. These trends are provided in Appendix E. Note that the performance of BCQ often tracks the performance of the BC baseline, suggesting that BCQ primarily imitates the data. Our KL-control baseline uses automatic temperature tuning [18]. We find that KL-control usually performs similar or worse to BC, whereas DQfD tends to diverge often due to cumulative error due to OOD actions and often exhibits a huge variance across different runs (for example, HalfCheetah-v2 environment).

### 6.2   Performance on Random and Optimal Datasets

In Figure 5, we show the performance of each method when trained on data from a random policy (top) and a near-optimal policy (bottom). In both cases, our method BEAR achieves good results, consistently exceeding the average dataset return on random data, and matching the optimal policy return on optimal data. Naïve RL also often does well on random data. For a random data policy, all actions are in-distribution, since they all have equal probability. This is consistent with our hypothesis

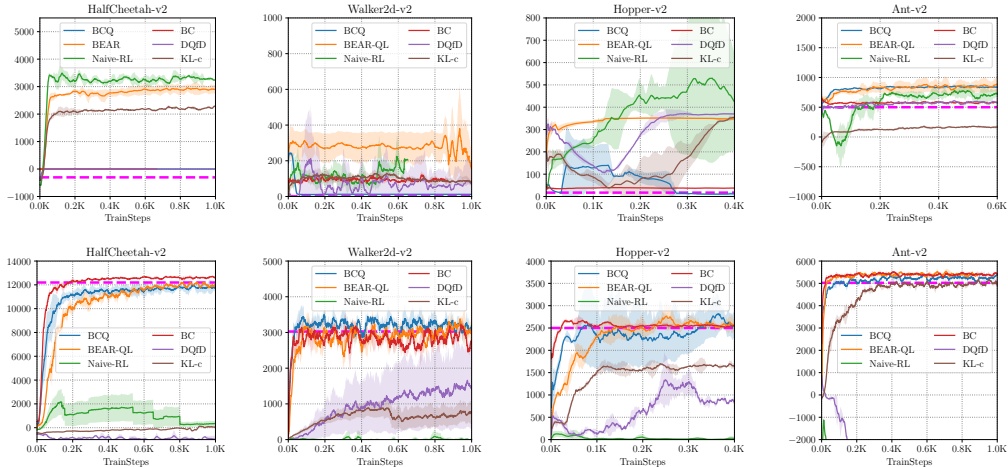

Figure 5: Average performance of BEAR-QL, BCQ, Naïve RL and BC on random data (top row) and optimal data (bottom row) over 5 seeds. BEAR-QL is the only algorithm capable of learning in both scenarios. Naïve RL cannot handle optimal data, since it does not illustrate mistakes, and BCQ favors a behavioral cloning strategy (performs quite close to behaviour cloning in most cases), causing it to fail on random data. Average return over the training dataset is indicated by the dashed magenta line.

that OOD actions are one of the main sources of error in off-policy learning on static datasets. The prior BCQ method [12] performs well on optimal data but performs poorly on random data, where the constraint is too strict. These results show that BEAR-QL is robust to the dataset composition, and can learn consistently in a variety of settings. We find that KL-control and DQfD can be unstable in these settings.

Finally, in Figure 4, we show that BEAR outperforms other considered prior methods in the challenging Humanoid-v2 environment as well, in two cases – Medium-quality data and random data.

## 6.3 Analysis of BEAR-QL

In this section, we aim to analyze different components of our method via an ablation study. Our first ablation studies the support constraint discussed in Section 5, which uses MMD to measure support. We replace it with a more standard KL-divergence distribution constraint, which measures similarity in density. Our hypothesis is that this should provide a more conservative constraint, since matching distributions is not necessary for matching support. KL-divergence performs well in some cases, such as with optimal data, but as shown in Figure 6, it performs worse than MMD on medium-quality data. Even

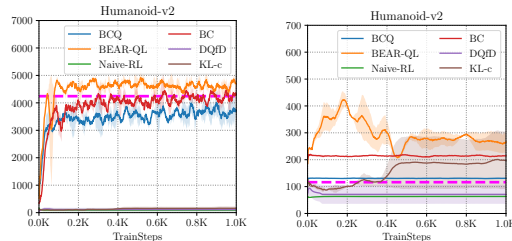

Figure 4: Performance of BEAR-QL, BCQ, Naïve RL and BC on medium-quality (left) and random (right) data in the Humanoid-v2 environment. Note that BEAR-QL outperforms prior methods.

when KL-divergence is hand tuned fully, so as to prevent instability issues it still performs worse than a not-well tuned MMD constraint. We provide the results for this setting in the Appendix. We also vary the number of samples $n$ that are used to compute the MMD constraint. We find that smaller n ($\approx$ 4 or 5) gives better performance. Although the difference is not large, consistently better performance with 4 samples leans in favour of our hypothesis that an intermediate number of samples works well for support matching, and hence is less restrictive.

## 7 Discussion and Future Work

The goal in our work was to study off-policy reinforcement learning with static datasets. We theoretically and empirically analyze how error propagates in off-policy RL due to the use of out-of-distribution actions for computing the target values in the Bellman backup. Our experiments suggest that this source of error is one of the primary issues afflicting off-policy RL: increasing the number

of samples does not appear to mitigate the degradation issue (Figure 1), and training with naïve RL on data from a random policy, where there are no out-of-distribution actions, shows much less degradation than training on data from more focused policies (Figure 5). Armed with this insight, we develop a method for mitigating the effect of out-of-distribution actions, which we call BEAR-QL. BEAR-QL constrains the backup to use actions that have non-negligible support under the data distribution, but without being overly conservative in constraining the learned policy. We observe experimentally that BEAR-QL achieves good performance across a range of tasks, and across a range of dataset compositions, learning well on random, medium-quality, and expert data.

While BEAR-QL substantially stabilizes off-policy RL, we believe that this problem merits further study. One limitation of our current method is that, although the learned policies are more performant than those acquired with naïve RL, performance sometimes still tends to degrade for long learning runs. An exciting direction for future work would be to develop an early stopping condition for RL, perhaps by generalizing the notion of validation error to reinforcement learning. A limitation of approaches that perform constrained-action selection is that they can be overly conservative when compared to methods that constrain state-distributions directly, especially with datasets collected from mixtures of policies. We leave

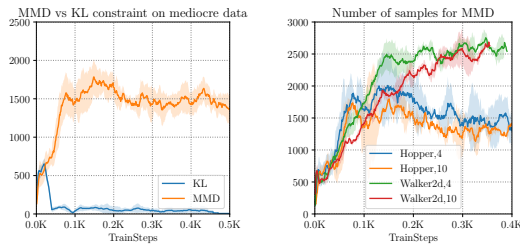

Figure 6: Average return (averaged Hopper-v2 and Walker2d-v2) as a function of train steps for ablation studies from Section 6.3. (a) MMD constrained optimization is more stable and leads to better returns, (b) 4 sample MMD is more performant than 10.

it to future work to design algorithms that can directly constrain state distributions. A theoretically robust method for support matching efficiently in high-dimensional continuous action spaces is a question for future research. Perhaps methods from outside RL, predominantly used in domain adaptation, such as using asymmetric f-divergences [37] can be used for support restriction. Another promising future direction is to examine how well BEAR-QL can work on large-scale off-policy learning problems, of the sort that are likely to arise in domains such as robotics, autonomous driving, operations research, and commerce. If RL algorithms can learn effectively from large-scale off-policy datasets, reinforcement learning can become a truly data-driven discipline, benefiting from the same advantage in generalization that has been seen in recent years in supervised learning fields, where large datasets have enabled rapid progress in terms of accuracy and generalization [7].

## Acknowledgements

We thank Kristian Hartikainen for sharing implementations of RL algorithms and for help in debugging certain issues. We thank Matthew Soh for help in setting up environments. We thank Aurick Zhou, Chelsea Finn, Abhishek Gupta and Kelvin Xu for informative discussions. We thank Ofir Nachum for comments on an earlier draft of this paper. We thank Google, NVIDIA, and Amazon for providing computational resources. This research was supported by Berkeley DeepDrive, JPMorgan Chase & Co., NSF IIS-1651843 and IIS-1614653, the DARPA Assured Autonomy program, and ARL DCIST CRA W911NF-17-2-0181.

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
