[Supplementary Material]

# Appendices

## A  Distribution-Constrained Backup Operator

In this section, we analyze properties of the constrained Bellman backup operator, defined as:

$$\mathcal{T}^{\Pi}Q(s,a) \stackrel{\text{def}}{=} \mathbb{E}\big[R(s,a) + \gamma \max_{\pi \in \Pi} \mathbb{E}_{P(s'|s,a)}\left[V(s')\right]\big]$$

where

$$V(s) \stackrel{\text{def}}{=} \max_{\pi \in \Pi} \mathbb{E}_{\pi}[Q(s,a)].$$

Such an operator can be reduced to a standard Bellman backup in a modified MDP. We can construct an MDP $M'$ from the original MDP $M$ as follows:

- The state space, discount, and initial state distributions remain unchanged from $M$.
- We define a new action set $\mathcal{A}' = \Pi$ to be the choice of policy $\pi$ to execute.
- We define the new transition distribution $p'$ as taking one step under the chosen policy $\pi$ to execute and one step under the original dynamics $p$: $p'(s'|s,\pi) = E_{\pi}[p(s'|s,a)]$.
- Q-values in this new MDP, $Q^{\Pi}(s,\pi)$ would, in words, correspond to executing policy $\pi$ for one step and executing the policy which maximizes the future discounted value function in the original MDP $M$ thereafter.

Under this redefinition, the Bellman operator $\mathcal{T}^{\Pi}$ is mathematically the same operation as the Bellman operator under $M'$. Thus, standard results from MDP theory carry over - i.e. the existence of a fixed point and convergence of repeated application of $\mathcal{T}^{\Pi}$ to said fixed point.

## B  Error Propagation

In this section, we provide proofs for Theorem 4.1 and Theorem 4.2.

**Theorem B.1.** *Suppose we run approximate distribution-constrained value iteration with a set constrained backup $\mathcal{T}^{\Pi}$. Assume that $\delta(s,a) \geq \max_k |Q_k(s,a) - \mathcal{T}^{\Pi}Q_{k-1}(s,a)|$ bounds the Bellman error. Then,*

$$\lim_{k \to \infty} \mathbb{E}_{\rho_0}[|V_k(s) - V^*(s)|] \leq \frac{\gamma}{(1-\gamma)^2}\left[C(\Pi)\mathbb{E}_{\mu}[\max_{\pi \in \Pi}\mathbb{E}_{\pi}[\delta(s,a)]] + \frac{1-\gamma}{\gamma}\alpha(\Pi)\right]$$

*Proof.* We first begin by introducing $V^{\Pi}$, the fixed point of $\mathcal{T}^{\Pi}$. By the triangle inequality, we have:

$$\mathbb{E}_{\rho_0}[|V_k(s) - V^*(s)|] = \mathbb{E}_{\rho_0}[|V_k(s,a) - V^{\Pi}(s) + V^{\Pi}(s) - V^*(s)|]$$

$$\leq \underbrace{\mathbb{E}_{\rho_0}[|V_k(s) - V^{\Pi}(s)|]}_{L_1} + \underbrace{\mathbb{E}_{\rho_0}[|V^{\Pi}(s) - V^*(s)|]}_{L_2}$$

First, we note that $\max_{\pi}\mathbb{E}_{\pi}[\delta(s,a)]$ provides an upper bound on the value error:

$$|V_k(s) - \mathcal{T}^{\Pi}V_{k-1}(s)| = |\max_{\pi}\mathbb{E}_{\pi}[Q_k(s,a)] - \max_{\pi}\mathbb{E}_{\pi}[\mathcal{T}^{\pi}Q_{k-1}(s,a)]|$$

$$\leq \max_{\pi}\mathbb{E}_{\pi}[|Q_k(s,a) - \mathcal{T}^{\pi}Q_{k-1}(s,a)|]$$

$$\leq \max_{\pi}\mathbb{E}_{\pi}[\delta(s,a)]$$

We can bound $L_1$ with

$$L_1 \leq \frac{2\gamma}{(1-\gamma)^2}[C(\Pi)]\mathbb{E}_{\mu}[\max_{\pi \in \Pi}\mathbb{E}_{\pi}[\delta(s,a)]]$$

by direct modification of the proof of Theorem 3 of Farahmand et al. [10] or Theorem 1 of Munos [28] with $k = 1$ ($p = 1$), but replacing $V^*$ with $V^{\Pi}$ and $\mathcal{T}$ with $\mathcal{T}^{\Pi}$, as $\mathcal{T}^{\Pi}$ is a contraction and

$V^\Pi$ is its fixed point. An alternative proof involves viewing $\mathcal{T}^\Pi$ as a backup under a modified MDP (see Appendix A), and directly apply Theorem 1 of Munos [28] under this modified MDP. A similar bound also holds true for value iteration with the $\mathcal{T}^\Pi$ operator which can be analysed on similar lines as the above proof and Munos [28].

To bound $L_2$, we provide a simple $\ell_\infty$-norm bound, although we could in principle apply techniques used to bound $L_1$ to get a tighter distribution-based bound.

$$
\begin{aligned}
\left\| V^\Pi - V^* \right\|_\infty &= \left\| \mathcal{T}^\Pi V^\Pi - \mathcal{T} V^* \right\|_\infty \\
&\leq \left\| \mathcal{T}^\Pi V^\Pi - \mathcal{T}^\Pi V^* \right\|_\infty + \left\| \mathcal{T}^\Pi V^\Pi - \mathcal{T} V^* \right\|_\infty \\
&\leq \gamma \left\| V^\Pi - V^* \right\|_\infty + \alpha(\Pi)
\end{aligned}
$$

Thus, we have $\left\| V^\Pi - V^* \right\|_\infty \leq \frac{\alpha}{1-\gamma}$. Because the maximum is greater than the expectation, $L_2 = \mathbb{E}_{\rho_0, \pi}[|V^\Pi(s) - V^*(s)|] \leq \left\| V^\Pi - V^* \right\|_\infty$.

Adding $L_1$ and $L_2$ completes the proof. $\qquad\square$

**Theorem B.2.** *Assume the data distribution $\mu$ is generated by a behavior policy $\beta$, such that $\mu(s,a) = \mu_\beta(s,a)$. Let $\mu(s)$ be the marginal state distribution under the data distribution. Let us define $\Pi_\epsilon = \{\pi \mid \pi(a|s) = 0 \text{ whenever } \beta(a|s) < \epsilon\}$. Then, there exists a concentrability coefficient $C(\Pi_\epsilon)$ is bounded as:*

$$
C(\Pi_\epsilon) \leq C(\beta) \cdot \left( 1 + \frac{\gamma}{(1-\gamma)f(\epsilon)}(1-\epsilon) \right)
$$

*where $f(\epsilon) \stackrel{\text{def}}{=} \min_{s \in \mathcal{S}, \mu_\Pi(s) > 0}[\mu(s)]$.*

*Proof.* For notational clarity, we refer to $\Pi_\epsilon$ as $\Pi$ in this proof. The term $\mu_\Pi$ is the highest discounted marginal state distribution starting from the initial state distribution $\rho$ and following policies $\pi \in \Pi$. Formally, it is defined as:

$$
\mu_\Pi \stackrel{\text{def}}{=} \max_{\{\pi_i\}_i: \forall i, \pi_i \in \Pi} (1-\gamma) \sum_{m=1}^{\infty} m\gamma^{m-1} \rho_0 P^{\pi_1} \cdots P^{\pi_m}
$$

Now, we begin the proof of the theorem. We first note, from the definition of $\Pi$, $\forall s \in \mathcal{S} \ \forall \pi \in \Pi, \pi(a|s) > 0 \implies \beta(a|s) > \epsilon$. This suggests a bound on the total variation distance between $\beta$ and any $\pi \in \Pi$ for all $s \in \mathcal{S}$, $D_{TV}(\beta(\cdot|s)\|\pi(\cdot|s)) \leq 1 - \epsilon$. This means that the marginal state distributions of $\beta$ and $\Pi$, are bounded in total variation distance by: $D_{TV}(\mu_\beta\|\mu_\Pi) \leq \frac{\gamma}{1-\gamma}(1-\epsilon)$, where $\mu_\Pi$ is the marginal state distribution as defined above. This can be derived from Schulman et al. [34], Appendix B, which bounds the difference in returns of two policies by showing the state marginals between two policies are bounded if their total variation distance is bounded.

Further, the definition of the set of policies $\Pi$ implies that $\forall s \in \mathcal{S}, \mu_\Pi(s) > 0 \implies \mu_\beta(s) \geq f(\epsilon)$, where $f(\epsilon) > 0$ is a constant that depends on $\epsilon$ and captures the minimum visitation probability of a state $s \in \mathcal{S}$ when rollouts are executed from the initial state distribution $\rho$ while executing the behaviour policy $\beta(a|s)$, under the constraint that only actions with $\beta(a|s) \geq \epsilon$ are selected for execution in the environment. Combining it with the total variation divergence bound, $\max_s \|\mu_\beta(s) - \mu_\Pi(s)\| \leq \frac{\gamma}{1-\gamma}(1-\epsilon)$, we get that

$$
\sup_{s \in \mathcal{S}} \frac{\mu_\Pi(s)}{\mu_\beta(s)} \leq 1 + \frac{\gamma}{(1-\gamma)f(\epsilon)}(1-\epsilon)
$$

We know that, $C(\Pi) \stackrel{\text{def}}{=} (1-\gamma)^2 \sum_{k=1}^{\infty} k\gamma^{k-1} c(k)$ is the ratio of the marginal state visitation distribution under the policy iterates when performing backups using the distribution-constrained operator and the data distribution $\mu = \mu_\beta$. Therefore,

$$
\frac{C(\Pi_\epsilon)}{C(\beta)} \stackrel{\text{def}}{=} \sup_{s \in \mathcal{S}} \frac{\mu_\Pi(s)}{\mu_\beta(s)} \leq 1 + \frac{\gamma}{(1-\gamma)f(\epsilon)}(1-\epsilon)
$$

$\qquad\square$

# C  Additional Details Regarding BEAR-QL

In this appendix, we address several remaining points regarding the support matching formulation of BEAR-QL, and further discuss its connections to prior work.

## C.1  Why can we choose actions from $\Pi_\epsilon$, the support of the training distribution, and need not restrict action selection to the policy distribution?

In Section 4.1, we designed a new distribution-constrained backup and analyzed its properties from an error propagation perspective. Theorems 4.1 and 4.2 tell us that, if the maximum projection error on all actions within the support of the train distribution is bounded, then the worst-case error incurred is also bounded. That is, we have a bound on $\max_{\pi \in \Pi_\epsilon} \mathbb{E}_\pi[\delta_k(s,a)]$. In this section, we provide an intuitive explanation for why action distributions that are very different from the training policy distributions, but still lie in the support of the train distribution, can be chosen without incurring large error. In practice, we use powerful function approximators for Q-learning, such as deep neural networks. That is, $\delta_k(s,a)$ is the Bellman error for one iteration of Q-iteration/Q-learning, which can essentially be viewed as a supervised regression problem with a very expressive function class. In this scenario, we expect a bounded error on the entire support of the training distribution, and we therefore expect approximation error to depend less on the specific density of a datapoint under the data distribution, and more on whether or not that datapoint is within the support of the data distribution. I.e., any point that is within the support would have a comparatively low error, due to the expressivity of the function approximator.

Another justification is that, a different version of the Bellman error objective renormalizes the action-distributions to the uniform distribution by applying an inverse behavior policy density weighting. For example, [2, 1] use this variant of Bellman error:

$$Q_{k+1} = \operatorname{argmin}_Q \sum_{i=1, a_i \sim \beta(\cdot|s_i)}^{N} \frac{1}{\beta(a_i|s_i)} \left( Q(s_i, a_i) - \left[ R(s,a) + \gamma \max_{a' \in \mathcal{A}} Q_k(s_{i+1}, a') \right] \right)^2$$

This implies that this form of Bellman error mainly depends upon the support of the behaviour policy $\beta$ (i.e. the set of action samples sampled from $\beta$ with a high-enough probability which we formally refer to as $\beta(a|s) \geq \epsilon$ in the main text). In a scenario when this form of Bellman error is being minimized, $\delta_k(s,a)$ is defined as

$$\delta_k(s,a) = \frac{1}{\beta(a|s)} |Q_k(s,a) - \mathcal{T}Q_{k-1}(s,a)|$$

The overall error, hence, incurred due to error propagation is expected to be insensitive to distribution change, provided the support of the distribution doesn't change. Therefore, all policies $\pi \in \Pi_\epsilon$ incur the same amount of propagated error ($|V_k - V_\Pi|$) whereas different amount of suoptimality biases – suggesting the existence of a different policy in $\Pi_\epsilon$ which propagates the same amount of error while having a lower suboptimality bias. However, in practice, it has been observed that using the inverse density weighting under the behaviour policy doesn't lead to substantially better performance for vanilla RL (not in the setting with purely off-policy, static datasets), so we use the unmodified Bellman error objective.

Both of these justifications indicate that bounded $\delta_k(s,a)$ is reasonable to expect under in-support action distributions.

## C.2  Details on connection between BEAR-QL and distribution-constrained backups

Distribution-constrained backups perform maximization over a set of policies $\Pi_\epsilon$ which is defined as the set of policies that share the support with the behaviour policy. In the BEAR-QL algorithm, $\pi_\phi$ is maximized towards maximizing the expected Q-value for each state under the action distribution defined by it, while staying in-support (through the MMD constraint). The maximization step biases $\pi_\phi$ towards the in-support actions which maximize the current Q-value. By sampling multiple Dirac-delta action distributions - $\delta_{a_i}$ - and then performing an explicit maximum over them for computing the target is a stochastic approximation to the distribution-constrained operator. What is the importance of training the actor to maximize the expected Q-value? We found empirically that this step is important as without this maximization step and high-dimensional action spaces, it

is likely to require many more samples (exponentially more, due to curse of dimensionality) to get the correct action that maximizes the target value while being in-support. This is hard and unlikely, and in some experiments we tried with this variant, we found it to lead to suboptimal solutions. At evaluation time, we use the Q-function as the actor. The same process is followed. Dirac-delta action distribution candidates $\delta_{a_i}$ are sampled, and then the action $a_i$ that is gives the empirical maximum over the Q-function values is the action that is executed in the environment.

### C.3  How effective is the MMD constraint in constraining supports of distributions?

In Section 5, we argued in favour of the usage of the sampled MMD distance between distributions to search for a policy that is supported on the same support as the train distribution. Revisiting the argument, in this section, we argue, via numerical simulations, the effectiveness of the MMD distance between two probability distributions in constraining the support of the distribution being learned, without constraining the distribution density function too much. While, MMD distance computed exactly between two distribution functions will match distributions exactly and that explains its applicability in 2-sample tests, however, with a limited number of samples, we empirically find that the values of the MMD distance computed using samples from two $d$-dimensional Gaussian distributions with diagonal covariance matrices: $P \stackrel{\text{def}}{=} \mathcal{N}(\mu_P, \Sigma_P)$ and $Q \stackrel{\text{def}}{=} \mathcal{N}(\mu_Q, \Sigma_Q)$ is roughly equal to the MMD distance computed using samples from $\mathcal{U}_\alpha(P) \stackrel{\text{def}}{=} [\,\text{Uniform}(\mu_P^1 \pm \alpha\Sigma_P^{1,1})\,] \times \cdots \times [\,\text{Uniform}(\mu_P^d \pm \alpha\Sigma_P^{d,d})\,]$ and $Q$. This means that when minimizing the MMD distance to train distribution $Q$, the gradient signal would push $Q$ towards a uniform distribution supported on $P$'s support as this solution exhibits a lower MMD value – which is the objective we are optimizing.

Figure 7 shows an empirical comparison of $\text{MMD}(P, Q)$ when $Q = P$, computed by sampling $n$-samples from $P$, and $\text{MMD}(\mathcal{U}_\alpha(P), Q)$ (also when $Q = P$) computed by sampling $n$-samples from $\mathcal{U}_\alpha(P)$. We observe that MMD distance computed using limited samples can, in fact, be higher between a distribution and itself as compared to a uniform distribution over a distribution's support and itself. In Figure 7, note that for smaller values of $n$ and appropriately chosen $\alpha$ (mentioned against each figure, the support of the uniform distribution), the estimator for $\text{MMD}(\mathcal{U}_\alpha(P), P)$ can provide lower estimates than the value of the estimator for $\text{MMD}(P, P)$. This observation suggests that when the number of samples is not enough to sample infer the distribution shape, density-agnostic distances like MMD can be used as optimization objectives to push distributions to match supports. Subfigures (c) and (d) shows the increase in MMD distance as the support of the uniform distribution is expanded.

In order to provide a theoretical example, we refer to Example 1 in Gretton et al. [17], and extend it. First, note that the example argues that a fixed sample size of samples drawn from a distribution $P$, there exists another discrete distribution $Q$ supported on $m^2$ samples from the support set of $P$, such that there atleast is a probability $\binom{m^2}{m} \frac{m!}{m^{2m}} > 1 - e^{-1} > 0.63$ that a sample from $Q$ is indeed a sample from $P$ as well. So, with a smaller value of $m$, *no* 2-sample test will be able to distinguish between $P$ and $Q$. We would also note that this example is exactly the argument that our algorithm build upon. We further extend this example by noting that if $Q$ were rather not completely supported on the support of $P$, then there exists atleast a probability of $\epsilon$ that a sample from $Q$ lies outside the support of $P$. This gives us a lower bound on the value of the MMD estimator, indicating that the MMD 2-sample test will be able to detect this distribution due to an irreducible difference of $\epsilon \sqrt{\min_{y \in \text{Extremal(P)}} \mathbb{E}_{x \sim P}[k(x, y)]}$ (where $y$ is an "extremal point" in $P$'s support) in the MMD estimate.

## D   Additional Experimental Details

**Data collection**   We trained behaviour policies using the Soft Actor-Critic algorithm [18]. In all cases, random data was generated by running a uniform at random policy in the environment. Optimal data was generated by training SAC agents in all 4 domains until convergence to the returns mentioned in Figure 5. Mediocre data was generated by training a policy until the return value marked in each of the plots in Figure 3. Each of our datasets contained 1e6 samples. We used the same datasets for evaluating different algorithms to maintain uniformity across results.

(a) $\mathcal{N}(0, 0.1), \mathcal{U}(-0.1, 0.1)$

(b) $\mathcal{N}(0, 1.0), \mathcal{U}(-1.5, 1.5)$

(c) $\mathcal{N}(0, 1.0), \mathcal{U}(-2.0, 2.0)$

(d) $\mathcal{N}(0, 1.0), \mathcal{U}(-4.0, 4.0)$

Figure 7: Comparing MMD distance between a 1-d Gaussian distribution ($P$) and itself ($P$), and a uniform distribution over support set of the $P$ and the distribution $P$. The parameters of the Gaussian distribution ($P$) and the uniform distribution being considered are mentioned against each plot. ('Self' refers to $\mathrm{MMD}(P, P)$ and 'Uniform' refers to $\mathrm{MMD}(P, \mathcal{U}(P))$.) Note that for small values of $n \approx 1 - 10$, the MMD with the Uniform distribution is slightly lower in magnitude than the MMD between the distribution $P$ and itself (sub-figures (a), (b) and (c)). For (d), as the support of this uniform distribution is enlarged, this leads to an increase in the value of MMD in the uniform approximation case – which suggests that a near-local minimizer for the MMD distance can be obtained by making sure that the distribution which is being trained in this process shares the same support as the other given distribution.

**Choice of kernels**  In our experiments, we found that the choice of the kernel is an important design decision that needs to be made. In general, we found that a Laplacian kernel $k(x, y) = \exp(\frac{-||x-y||}{\sigma})$ worked well in all cases. Gaussian kernel $k(x, y) = \exp(\frac{-||x-y||^2}{2\sigma^2})$ worked quite well in the case of optimal dataset. For the Laplacian kernel, we chose $\sigma = 10.0$ for Cheetah, Ant and Hopper, and $\sigma = 20.0$ for Walker. However, we found that $\sigma = 20.0$ worked well for all environments in all settings. For the Gaussian kernel, we chose $\sigma = 20.0$ for all settings. Kernels often tend to not provide relevant measurements of distance especially in high-dimensional spaces, so one direction for future work is to design right kernels. We further experimented with a mixture of Laplacian kernel with different bandwidth parameters $\sigma$ $(1.0, 10.0, 50.0)$ on Hopper-v2 and Walker2d-v2 where we found that it performs comparably and sometimes is better than a simple Laplacian kernel, probably because it is able to track supports upto different levels of thresholds due to multiple kernels.

**More details about the algorithm**  At evaluation time, we find that using the greedy maximum of the Q-function over the support set of the behaviour policy (which can be approximated by sampling multiple Dirac-delta policies $\delta_{a_i}$ from the policy $\pi_\phi$ and performing a greedy maximization of the Q-values over these Dirac-delta policies) works best, better than unrolling the learned actor $\pi_\phi$ in the environment. This was also found useful in [12]. Another detail about the algorithm is deciding which samples to use for computing the MMD objective. We train a parameteric model $\pi_{data}$ which fits a tanh-Gaussian distribution to $a$ given the states $s$, $\pi_{data}(\cdot|s) = \tanh \mathcal{N}(\mu(\cdot|s), \sigma(\cdot|s))$ and then use this to sample a candidate $n$ actions for computing the MMD-distance, meaning that MMD is computed between $a_1, \cdots, a_N \sim \pi_{data}$ and $\pi_\phi$. We find the latter to work better in practice. Also, computing the MMD distance between actions before applying the tanh transformation work better, and leads to a constraint, that perhaps provides stronger gradient signal – because tanh saturates very quickly, after which gradients almost vanish.

**Other hyperparameters**  Other hyperparameters include the following – (1) The variance of the Gaussian $\sigma^2$ /(standard deviation of) Laplacian kernel $\sigma$: We tried a variance of 10, 20, and 40. We found that 10 and 20 worked well across Cheetah, Hopper and Ant, and 20 worked well for Walker2d; (2) The learning rate for the Lagrange multiplier was chosen to be 1e-3, and the $\log$ of the Lagrange

multiplier was clipped between $[-5, 10]$ to prevent instabilities; (3) For the policy improvement step, we found using average Q works better than min Q for Walker2d. For the baselines, we used BCQ code from the official implementation accompanying [12], TD3 code from the official implementation accompanying [13] and the BC baseline was the VAE-based behaviour cloning baseline also used in [12]. We evaluated on 10 evaluation episodes (which were separate from the train distribution) after every 1000 iterations and used the average score and the variance for the plots.

# E  Additional Experimental Results

Figure 8: The trend of the difference between the Q-values and Monte-Carlo returns: $Q - MC$ returns for 2 environments. Note that a high value of $Q - MC$ corresponds to more overestimation. In these plots, BEAR-QL is more well behaved than BCQ. In Walker2d-v2, BCQ tends to diverge in the negative direction. In the case of Ant-v2, although roughly the same, the difference between Q values and Monte-carlo returns is slightly lower in the case of BEAR-QL suggestion no risk of overestimation. (This corresponds to medium-quality data.)

Figure 9: The trends of Q-values as a function of number of gradient steps taken in case of 3 environments. BCQs Q-values tend to be more unstable (especially in the case of Walker2d, where they diverge in the negative direction) as compared to BEAR-QL. This corresponds to medium-quality data.

In this section, we provide some extra plots for some extra experiments. In Figure 8 we provide the difference between learned Q-values and Monte carlo returns of the policy in the environment. In Figure 9 we provide the trends of comparisons of Q-values learned by BEAR-QL and BCQ in three environments. In Figure 10 we compare the performance when using the MMD constraint vs using the KL constraint in the case of three environments. In order to be fair at comparing to MMD, we train a model for the behaviour policy and constrain the KL-divergence to this behaviour policy. (For MMD, we compute MMD using samples from the model of the behaviour policy.) Note that in the case of Half Cheetah with medium-quality data, KL divergence constraint works pretty well, but it fails drastically in the case of Hopper and Walker2d and the Q-values tend to diverge. Figure 10 summarizes the trends for 3 environments.

We further study the performance of the KL-divergence in the setting when the KL-divergence is stable. In this setting we needed to perform extensive hyperparameter tuning to find the optimal

Figure 10: Performance Trends (measured in AverageReturn) for Hopper-v2, HalfCheetah-v2 and Walker2d-v2 environments with BEAR-QL algorthm but varying kind of constraint. In general we find that using the KL constraint leads to worse performance. However, in some rare cases (for example, HalfCheetah-v2), the KL constraint learns faster. In general, we find that the KL-constraint often leads to diverging Q-values. This experiment corresponds to medium-quality data.

Figure 11: Performance Trends (measured in Average Returns) for Hopper-v2 and Walker2d-v2 environments with BEAR-QL algorithm with an extensively tuned KL-constraint and the MMD-constraint from. Note that the MMD-constraint still outperforms the KL-constraint.

Lagrange multiplier for the KL-constraint and plain and simple dual descent always gave us an unstable solution with the KL-constraint. Even in this case tuned hyperparameter case, we find that using a KL-constraint is worse than using a MMD-constraint. Trends are summarized in Figure 11.

As described in Section C, we can achieve a reduced overall error $||V_k(s) - V^*(s)||$, if we use the MMD support-matching constraint alongside importance sampling, i.e. when we multiply the Bellman error with the inverse of the behaviour policy density. Empirically, we tried reweighting the Bellman error by inverse of the fitted behavior policy density, alongside the BEAR-QL algorithm. The trends for two environments and medium-quality data are summarized in Figure 12. We found that reweighting the Bellman error wasn't that useful, although in theory, it provides an absolute error reduction as described by Theorem 4.1. We hypothesize that this could be due to the possible reason that when optimizing neural nets using stochastic gradient procedures, importance sampling isn't that beneficial [5].

Figure 12: BEAR with importance sampled Bellman error minimization. We find that importance sampling isn't that beneficial in practice.