[Reviews · NeurIPS 2019]

Reviewer 1



The proposal to restrict the learned policy to lie within the support of the data distribution instead of mimicking the data distribution is very good. Proposing this, and especially making it work is in my understanding novel (I will have to defer to the other reviewers who are hopefully deeper experts on Q-learning to comment on past work). The theoretical analysis that supports the idea is also well made: there is an appropriate balance between complexity and usefulness in the analysis. The empirical results are convincing and support the theory, and are in line with what one would expect. Overall, the paper seems solid and well written, the method relevant, and the results are good. Edit: Thank you for your response, and for adding the humanoid experiments, they add value to the paper. Although I am happy with the response and believe the paper got even stronger, I would say that my score pre- and post-update would round to the same fairly favorable number.

Reviewer 2



Summary: This paper proposes a new algorithm that help stabilize off-policy Q-learning. The idea is to introduce approximate Bellman updates that are based on constraint actions sampled only from the support of the training data distribution. The paper shows the main source of instability is the boostrapping error. The boostrapping process might use actions that do not lie in the training data distribution. This work shows a way to mitigate this issue. It provides both theoretical results and a practical algorithm. The experiment show some aspects of the idea. In overall, the paper studies an interesting problem in RL. The paper comes with many theoretical results, the practical algorihtm seems to work with promising experiment results. However the writing is quite hard to follow. The notations and explanations are not good enough to make it easily readable. Both the idea of using constraint actions for approximate Bellman updates and the theoretical results make sense. The paper looks like a premature work and needs substantial improvement to become a solid and stronger paper. Some of my major comments are: * Background section: - inconsistent notation: P(s'|s,a), T(s'|s,a) , p(s'|s,a); P^{\pi}, p_{\pi} - This section should at least include a description of SAC and TD3, before discussing about it in Section 4, especially in the example. * Section 4 & 5: - Example 1: the description is quite unclear. It's hard to understand the task and results. Especially the results here use SAC as a baseline, but all previous discussion use the standard Q-learning algorithm. - These results in Section 4 still lack intuition and especially connections to the proposed practical algorithm, BEAR. The algorithm BEAR uses many approximate techniques, e.g. Dirac policies, and the conservative estimate of Q. All these technical steps do not come with good explanation, especially the policy improvement step, e.g why estimate's variance is used, \Pi_\epsilon, definition of \Delta, {\cal D}(s). - About the arguments from lines 127-138 The support is different from the behaviour distribution/policy? it does not clarify this difference. How to understand the set \Pi_\epsilon, does it include all policies that have probability of choosing a smaller than epsilon? Though the meaning of the MMD constraint in Eq.1 is consistent with some discussion in the paper that the policy should be selecting actions lying in the support of \beta distribution, but is this consistent with the definition of \Pi_\epsilon? In addition, this MMD constraint in Eq. 1 looks similar to the constraint used by BCQ, so the discussion in line 127-138 is quite misleading? I think this also wants to constraint the policy to be learned that somehow must be close to the behaviour policy? - No definitions for k, \var_k {Q_k} , and \var_k {E[Q_k]} in line 210,211 - The choice of conservative estimate for policy: what is the intuition behind this? it is not upper or lower confidence bound, which is a way of exploration bonus. - Algorithm 1: the sampling is a mini-batch, however the description of Q-update and Policy-update do not tell there is updates using mini-batch. The update of the ensembles on 4-6 is also not explained: why the next Q value y(s,a) is computed as in step 5, especially regarding to \lambda? * Section 6: I wonder how this should be compared to [1]? and should related work consist of discussions to methods like [1]? - The results in Fig. 3: It seems BC can also be very competitive, e.g. HalfCheetah and Ant. I wonder why the performance of BEAR becomes degenerate in Hopper? - Figure 4 shows BEAR might be competitive in average for both settings. That means BEAR is not favorable in any cases. Given optimal data, nay method can be competitive, while with random data, either Naive RL or BCQ is very competitive. [1] Todd Hester et. al. Deep Q-learning from Demonstrations ----------------- The rebuttal has addressed some of my concerns. The only concern is about clarity and novelty.

Reviewer 3



This work aims to resolve the off-policy problem. Authors discussed a variant of value iteration operator which is induced by a set of constrained policies. They further identified two kinds of errors and showed that a trade-off between them can improve the performance. Finally, they leveraged the intuition to develop a practical algorithm. Their observation is very interesting and the solution makes sense. Clarity: the bootstrapping error is a central notion in this work. However, authors didn't give a definition of bootstrapping error before discussing how to deal with the bootstrapping error in line 127. Though I understood their results through their theorem, I didn't figure out what is bootstrapping error exactly. Empirical evaluation: the proposed algorithm seems to work well in practice. But two baselines is not very sufficient, which hurts this work somehow.

[Author Response · NeurIPS 2019]

We thank the reviewers for their detailed comments. Our primary contribution is developing theoretical insights and a
practical deep RL algorithm (BEAR) for learning from static, off-policy datasets without interaction. The key idea in
BEAR is to learn the best policy *within the support of the behaviour/data distribution*.

We have revised the text for clarity, evaluated methods on a more complex task **(Humanoid-v2)** (as requested by
**(R1)**), and added two baselines **(R3)**: DQfD (Hester et.al., 2017) (with static data only, as requested by **R2**) and KL-
control (KL-c) (c.f., Jacques et.al., 2019). BEAR outperforms all methods on Humanoid-v2, and BEAR outperforms
DQfD/KL-c on all other benchmark tasks as well (a subset visualized below). We will release code with the final **(R1)**.

**R2: BEAR is not favourable**. BEAR is the only algorithm that achieves competitive performance across all dataset
compositions. Naive RL fails with optimal data, and BCQ/BC/DQfD/KL-c fail on random data. In practice, most
logged datasets are between optimal data and random data ("medium quality"), and BEAR outperforms all methods
(BCQ/BC/DQfD/KL-c/Naive RL), often by a *large* margin, in this setting (Fig 3, *orange line*; Figures above).

**R2: arguments from lines 127-138: restricting supports vs distributions; comparison to constraint in BCQ** We
have rewritten the paragraph to clarify the argument. BCQ implicitly constrains the learned policy $\pi(a|s)$ to be close
to the behaviour policy $\beta$. BEAR, on the other hand, relaxes the constraint to only enforce a support constraint,
that is $\pi(a|s)$ has positive density *only where* the density of the behaviour policy is more than a threshold (i.e.,
$\forall a, \beta(a|s) \leq \varepsilon \implies \pi(a|s) = 0$). $\Pi_\varepsilon \subseteq \Delta^{|S|}$ (where $\Delta$ denotes the simplex) is the set of policies, that satisfy this
support constraint. Our experiments (Fig 3; Figures above) show that this crucial difference allows BEAR to outperform
prior methods especially when the logged data is suboptimal.

**R2: Results in Sec 4 lack intuition and connection to BEAR** Theorem 4.1 shows a trade-off (lines 172-177) between
propagated error and suboptimality bias due to restricting the backups ($\alpha(\Pi)$). In practice, BEAR restricts $\pi_\phi$ to
lie in the support of $\beta$. This insight is formally justified in Theorems 4.1 & 4.2 ($C(\Pi_\varepsilon)$ is bounded). Computing
distribution-constrained backup exactly by maximizing over $\pi \in \Pi_\varepsilon$ is intractable in practice. As an approximation, we
sample Dirac policies in the support of $\beta$ (Alg 1, Line 5) and perform empirical maximization to compute the backup.
As the maximization is performed over a *narrower* set of Dirac policies ($\{\delta_{a_i}\} \subseteq \Pi_\varepsilon$), the bound in Theorem 4.1 still
holds. Empirically, we show this approximation is sufficient to outperform previous methods. This connection is briefly
described in Appendix C.2. We now include an explanation of this in Section 5.

**R2: How does MMD relate to $\Pi_\varepsilon$?** Directly using MMD would constrain the learned policy to be similar to the
behaviour policy in distribution. However, critically, we use a small number of samples to form a sampled MMD
estimate. In Appendix C.3, we show that sampled MMD computed from a small number of samples has the effect of
measuring support matching, while allowing the relative density on the support to vary. Hence, by penalizing sampled
MMD between $\pi$ and $\beta$, we approximately constrain $\pi$ to $\Pi_\varepsilon$. The number of samples used for the sampled MMD
maps to the threshold $\varepsilon$ in $\Pi_\varepsilon$. Further, for discrete distributions, Gretton et al. (2012)'s example can be adapted to
show that sampled MMD with few samples exhibits the desired behavior. We have added this discussion in Section 5.

**R1: Principled methods for support restriction** We are glad that the reviewer found using MMD for support
restriction neat. While this choice is partially justified due to reasons mentioned in Lines 29-32 in this rebuttal, a
theoretically robust method is an important next step. We have added a discussion of this in Section 5 as future work
and a (theoretical) limitation of our method.

**R2: conservative estimate using ensembles; why variance?** For the practical algorithm, we used a conservative
estimate of Q, as this mitigates overestimated Q-values and leads to improved performance (c.f., TD3). Theoretically,
the estimate arises as a high-confidence, lower bound on the true (expected) value via Cantelli's inequality and was used
previously with bandits (e.g., CRM, Swaminathan et al. 2015). We have added this intuition and motivation in Sec 5.

**R2: $\lambda$ combination for target values** BCQ introduced the idea of using a soft-min $\lambda Q_{min} + (1 - \lambda)Q_{max}$. As also
reported by BCQ, we find that it performs better than using $Q_{min}$ for the target. We have clarified this in main text.

**R2, R3: Baselines:** We have added two baselines (DQfD and KL-c), and will add additional baselines in the final if the
reviewer has further suggestions. Note, DQfD assumes *optimality* of the static data, which can degrade performance
when used with suboptimal data. DQfD, by default, performs online interaction as well.

[Meta-Review · NeurIPS 2019]

The reviewers were in consensus about the merits of this paper, in particular the value of the proposed approach and the theoretical analysis. Some concerns were raised about the experimental validation but these have been alleviated by the new results and baselines added during rebuttal. Some concerns remain regarding the clarity of the paper. The authors claim to have revised the text but we are not able to see it to validate that it has improved in this respect. The authors are strongly encouraged to put some real effort into improving the clarity of the final version. Overall a solid paper.